# Influence of Different Heat-Stress-Reducing Systems on Physiological and Behavioral Responses and Social Dominance of Holstein and Jersey Cows and Heifers on Pasture

**DOI:** 10.3390/ani12182318

**Published:** 2022-09-07

**Authors:** Karoline L. Guimarães-Yamada, Geraldo T. dos Santos, Jesus A. C. Osório, Micheli R. Sippert, Monique Figueiredo-Paludo, Bianca G. L. da Silva, Júlio C. Damasceno, Chaouki Benchaar

**Affiliations:** 1Department of Animal Sciences, State University of Maringá, Maringá 87020900, Brazil; 2INCT-LEITE/CNPq—Araucária Foundation, State University of Maringá, Maringá 87020900, Brazil; 3Sherbrooke Research and Development Centre, Agriculture and Agri-Food Canada, Sherbrooke, QC J1M 0C8, Canada

**Keywords:** behavior, cow, pasture, physiological thermal stress

## Abstract

**Simple Summary:**

A suitable environment for dairy cows has become increasingly important, especially in the summer in tropical countries. Excess heat increases body, hormonal, reproductive responses, and behavioral temperature, and may in extreme cases lead to death. This research shows the importance of using systems that can reduce the heat stress of dairy cattle raised on pasture. In this study, climatic parameters of the thermal state of the environment during the period, and the physiological, behavioral, and preferential responses of Holstein and Jersey cows and heifers on pasture were evaluated. The results showed that dairy cows preferred to use sprinklers or showers over artificial shade under ambient conditions, especially lactating and Holstein cows versus pubertal heifers because they are more thermosensitive. These systems are more efficient in dissipating heat by evaporation, improving the welfare of the animals, especially those raised on pasture.

**Abstract:**

High ambient temperatures and relative humidity affect the behavior and physiology of the animal. This study investigated the influence of different heat-stress-reducing systems on the physiological, behavioral, and preferential responses of Holstein and Jersey cows and heifers on pasture. Experimental treatments were: (1) three heat-stress-reducing systems (sprinklers + artificial shade; showers + artificial shade; and artificial shade); (2) two breeds (Holstein and Jersey); and (3) two physiological stages (lactating cows and pubertal heifers). Physiological and behavioral responses to treatments were measured every 30 min on collection days. The frequency and duration of the use of the systems were recorded continuously 24 h/day for 3 days in each period. The air temperature and the relative humidity were 26 ± 4.1 °C and 74 ± 11.3%, respectively. The experimental treatments affected (*p* = 0.0354) standing idle, grazing behavior (*p* = 0.0435), and the frequency and duration of use of the systems by the animals (*p* < 0.0001). For all treatments, the respiratory rate and the coat surface temperature were highly and significantly correlated (*p* < 0.05) with the temperature and humidity index. In conclusion, under ambient conditions, dairy cows preferred using sprinklers or showers over artificial shade. These systems were more efficient at reducing the heat load and led to better behavioral and physiological responses.

## 1. Introduction

High ambient temperatures, relative humidity, and solar radiation affect the behavior, physiology, and welfare of animals, especially in tropical countries during the summer [1]. The excess heat increases body temperature [2] and respiratory rate [3] and reduces food intake [4], milk production [5], and reproductive performance, and may in extreme cases lead to death [6].

To reduce overheating, mitigation measures can be taken, one of which is allowing access to shade [1]. Shade can be created using a variety of materials and has the purpose of reducing direct solar radiation, favoring a better microclimate [7]. An increase in air temperature motivates animals to use and compete for shaded places and such behavior is considered a valuable mechanism for reducing body temperature [8,9] but is often inefficient at reducing excess heat. The use of water to cool the animals is more efficient at reducing heat than shade [9] because water has a high latent heat of evaporation and when evaporating it takes with it a large amount of heat [10]. The outcomes following cooling with water include an increase in feed intake and milk yield [11], and a reduction in respiratory rate, body temperature, the number of insects, and the number of movements of hooves and tails [9].

The study of systems that use water to reduce thermal stress experienced by animals is long-standing, mainly in feedlot systems, where water is frequently used in the trough line. For example, Igono et al. [12] studied the use of sprinklers plus ventilation in the trough line on physiological and productive responses and concluded that animals that were sprinkled with more ventilation had lower rectal temperatures and produced 2 kg more milk per day when compared with animals with access to shade only. Tao et al. [13] tested the combination of spraying and ventilation versus no air conditioning and found a better rectal temperature (39 vs. 39.4 °C), respiratory rate (45.6 vs. 78.4 mov/min), and milk production (33.9 vs. 28.9 kg/day) for the group of cows subjected to spraying and ventilation. However, Schutz et al. [9] studied the behavioral preference of cows for sprinklers, shade, and the control environment (no shade or sprinklers) and found a preference for shade over sprinklers and the control environment; however, the authors reported that the best physiological parameters were found in the sprinkler system.

The use of water on animals on pasture is relatively new [6] and data are scarce. Therefore, our objectives were to examine the influence of different thermal stress reduction systems on pasture (sprinklers, showers, artificial shading, and their combinations) during the hot season, in a tropical country, and to evaluate the preference for and behavior of animals in these systems. We predicted that sprinklers and showers would be more efficient at reducing the thermal load and consequently would improve physiological, comfort, and welfare results and that lactating and Holstein animals would have a greater need to use the systems with water.

## 2. Materials and Methods

### 2.1. Site and Duration

The experiment was carried out from 10 October to 21 November 2018 (Southern Hemisphere spring) at Fazenda Experimental de Iguatemi (FEI), a property belonging to the State University of Maringá (UEM), located at latitude 23°25′ S, longitude 51°57′ W, and 550 m of altitude.

### 2.2. Weather

According to the Köppen classification, the climate at Fazenda Experimental de Iguatemi is humid subtropical mesothermal (Cfa) with hot summers and rainfall concentrated in the summer months. During the experimental period, there was no rain and the mean temperature was 26 °C ± 4.1 °C while the temperature and humidity index (THI) was 73.4 ± 5.68. The maximum and the minimum environmental temperature recorded were 39.7 °C and 15.4 °C, respectively. The maximum and the minimum THI were 83.9 and 61.4, respectively (Table 1).

### 2.3. Animals, Handling, and Feeding

Twelve animals were selected for this study: three Holstein multiparous cows (600 ± 30 kg, 53 ± 11 months of age, and average milk yield of 27 ± 3.5 kg of milk/day between the third and fourth month of lactation 105 ± 20 day); three multiparous Jersey cows (370 ± 11 kg, 40 ± 6 months of age, and average milk yield 11 ± 1.5 kg of milk/day between the fifth and seventh month of lactation 180 ± 30 days); three Holstein heifers (325 ± 25 kg, 16 ± 0.6 months of age); and three Jersey heifers (250 ± 25 kg and 13 ± 0.6 months of age). All animals were identified with ear tags and necklaces to facilitate observation from a distance and data collection. However, as in each paddock, there was only one animal from each Latin square (Holstein cows; Jersey cows; Holstein heifers; and Jersey heifers), and identification by the difference in sizes was easily accomplished.

The animals were kept in paddocks of *Cynodon plectostachyus* pasture (17% crude protein and 53% neutral detergent fiber) of similar areas (approximately 2780 m^2^), where the systems to be tested were installed. The lactating cows were milked twice daily, in a 4 × 1 herringbone milking parlor using a mechanical milking machine, at approximately 6 am and 3 pm, then fed with corn silage (8% crude protein and 50% neutral detergent fiber) and concentrate (26% crude protein and 10% neutral detergent fiber) and released back into the paddocks. The heifers were fed on pasture with silage (8% crude protein and 50% neutral detergent fiber) and total concentrate (25% crude protein and 15% neutral detergent fiber) while the cows were milked so that they did not interfere with the heifers’ feed intake. The amount of feed offered to animals was determined based on their dry matter intake (DMI) of the previous day and by keeping refusals between 10% and 20% of the total feed supplied. 

### 2.4. Design and Experimental Treatments

Animals were used in a replicated 3 × 3 Latin square design with a 3 × 2 × 2 factorial arrangement of treatments. Experimental treatments were (Figure 1): three heat-stress-reducing systems [sprinklers + artificial shade (Tspr) (Figure 1a); showers + artificial shade (Tsho) (Figure 1b); and artificial shade (Tas) (Figure 1c)]; (2) two breeds (Holstein and Jersey); and (3) two physiological stages (lactating cows and pubertal heifers). To give the animals time to become accustomed to the systems, a pre-adaptive phase was included in which troughs with feed were placed in front of each system to guide the animals to them and so encourage their usage. Each experimental period lasted 14 days. The 1st to the 11th days were for adaptation of the animals to the systems and groups, and the 12th to the 14th days were for sample collection.

All paddocks had the same artificial shading system at one end, comprising eucalyptus beams in which iron-based structures were fixed (3.5 m tall in total) with synthetic fabric (artificial shade) providing 80% solar retention, which guaranteed a shaded area of 16 m^2^ (4 × 4 m) (Figure 1d). The sprinkler systems and showers were installed in two of the paddocks on iron structures of 8 m^2^ (4 × 2 m) on a cement base. Access to sprinklers and showers was possible freely on all four sides of the structure. Both systems were powered by a 10,000-L water tank and a 2.5 horsepower electric water pump. When the animals passed or positioned themselves under the systems, Intelbrás presence sensors (model IVP 3000 PET) sent information to a relay (electromechanical switch) that opened the valves, releasing water to the sprinklers and showers. The sprinkler system had six volcano sprinklers with a flow rate of 8 L/minute. The shower system had six simple showers with a flow rate of 12 L/minute. When the animals left the sensors’ range, the relays closed the water valves, ceasing the release of water into the systems.

### 2.5. Environmental Conditions

Environmental conditions including air temperature (T; °C), relative humidity (RH; %), and solar radiation (R; W/m^2^) were recorded by the FEI weather station every half hour on the sample collection days. Using the data obtained, the temperature and humidity index (THI) was calculated based on the equation reported by Thom [14]:(1)THI=T+0.36×Tdp+41.2
where: *T*: air temperature (°C) and *Tdp*: dew point temperature (°C).

### 2.6. Physiological Parameters 

Physiological parameters such as respiratory rate (RR; breaths/min) and coat surface temperature (CS; °C) were measured every half hour on the sample collection days from 9:00 a.m. to 6:30 p.m., except at the time of milking, in all animals. The RR was obtained by counting the flank movements for one minute, and the coat surface temperature (CS; °C) was measured using a thermal camera (Fluke Ti100 Infrared Thermal Imaging Camera), taking a measurement approximately one meter away from the animals. The rectal temperature (RT; °C) was recorded manually at 9:00 a.m. and 3:00 p.m. using a digital thermometer inserted into the rectum of the animals.

### 2.7. Behavior

The behavioral patterns of each animal were analyzed every half hour on the sample collection days from 9:00 a.m. to 6:30 p.m., except at the time of milking, in all animals by trained observers who took notes on specific worksheets. Behaviors were classified as: grazing (Gra) (the animals were considered to be eating if feed grass was being ingested or could be seen in the mouth); ruminating (Rumi) (defined as chewing movements without fresh feed in the mouth, regurgitation of feed or both); standing idle (SI) (the animals were considered to be standing idle if standing without performing other activities); lying idle (LI) (the animals were considered to be lying idle if their flank was in contact with the ground); and grooming (Groo) (characterized by intentional contact by the animal with objects, itself, or another animal).

### 2.8. Frequency and Duration of Use of the Systems

The frequency and duration of use of the systems were recorded continuously 24 h/day for 3 days in each period using video cameras (HDCVI Intelbrás 720p) installed in each paddock. All cameras were connected to a digital video recorder with surveillance software (Intelbrás^®^ DVR 3016). 

The frequency of use of sprinkler and shower systems was defined as the activation of these systems in response to the animal being positioned within the range of the sensors for more than 1 s. The frequency of use of artificial shading was determined visually by trained evaluators during filming and required the animal to be in the shaded area for more than 1 s. The duration of use of the systems was measured by trained evaluators as the difference between entry to and exit from the systems, be they sprinklers, showers, or artificial shading.

The data obtained were analyzed in the following ways:

Systems: use of sprinkler + artificial shading, in paddock 1; shower + artificial shading, in paddock 2; and only artificial shading, in paddock 3.

Comparison between the structures of the same paddock: sprinkling vs. artificial shading, in paddock 1; shower vs. artificial shading, in paddock 2; and artificial shading in paddock 3.

Time of day: 9:00 a.m. to 11:00 a.m.; 12:00 p.m. to 3:00 p.m.; and 4:00 p.m. to 6:00 p.m.

### 2.9. Statistical Analysis

The data were analyzed using the MIXED procedure of the SAS (Statistical Analysis System, 9.3 - SAS Institute, Inc., Cary, NC, USA). When there were interactions between the factors, Fisher’s least significant difference test (LSD) was used to identify the interaction.

Behavior, duration of use, and frequency of use according to the following model:Yijklmn = μ + Si + *Aj:i* + Pk + Tl + BRm + AGn + T × BRlm + T × AGln + eijklmn(2)
where *p_k_* ≈ *N* (0, $\sigma_{p}^{2}$), *a_l_* ≈ *N* (0, $\sigma_{a}^{2}$) and *e_ijkl_* ≈ *N* (0, $\sigma_{e}^{2}$), and where Y_ijkl_ is the observed value; μ is the general mean; Si is the fixed effect of animal within a square; Aj:i is the random effect of the animal within each Latin square; Pk is the experimental period fixed effect; T_l_ is the treatment (l = 1 and 3) fixed effect; BR*m* is the breed effect (m = 1 and 2); AG_n_ is the age group (n = 1 and 2) fixed effect; T×BRlm is the interaction between treatment and breed fixed effect; T × AGln is the interaction between treatment and age group fixed effect; eijklmn is the residual error; N indicates a normal distribution; and $\sigma_{p}^{2}$, $\sigma_{a}^{2}$ and $\sigma_{e}^{2}$ are the variances associated with random effects associated with period and animal, and residual variance, respectively.

The duration of use and frequency of use data were according to the following model:Yijklmn = μ + Si + *Aj:i* + Pk+ Tl + Hm +T × Hlm + eijklm(3)
where *p_k_* ≈ *N* (0, $\sigma_{p}^{2}$), *a_l_* ≈ *N* (0, $\sigma_{a}^{2}$) and *e_ijkl_* ≈ *N* (0, $\sigma_{e}^{2}$), and Y_ijkl_ is the observed value; μ is the general mean; Si is the animal fixed effect within a square; Aj:i is the random effect of the animal within each Latin square; Pk is the experimental period fixed effect; T_l_ is the treatment (l = 1 and 3) fixed effect; H_l_ is the hours (n = 1 from 9 a.m. to 11 a.m., n = 2 from 12 to 3 p.m., n = 3 from 4 p.m. to 6 p.m.) fixed effect; T × H_il_ is the hours x treatment interaction; *P_m_* is the period (l = 1 and 2) fixed effect; *a_n_* is the animal fixed effect; *e_ijklm_* is the residual error; N indicates a normal distribution; and $\sigma_{p}^{2}$, $\sigma_{a}^{2}$ and $\sigma_{e}^{2}$ are the variances associated with random effects associated with period and animal, and residual variance, respectively. 

Comparisons between THI, respiratory rate, and coat temperature were made through linear regression adjustment and Pearson’s correlation coefficient (r). The adjustment adequacy was evaluated by the determination coefficient (R^2^). The level of significance adopted was 5% (*p* < 0.05).

## 3. Results

### 3.1. Environmental Conditions

The environmental conditions recorded during the experimental period are shown in Table 1. The air temperature was 26 ± 4.1 °C (mean ± standard deviation). The maximum environmental temperature recorded was 39.7 °C and the minimum temperature was 15.4 °C. The relative humidity (RH) during the experimental period was 74 ± 11.3%, on average. The maximum and the minimum RH were 94% and 41%, respectively. The average radiation was 338.2 ± 585.6 (W/m^2^) and the THI was 73.4 ± 5.68. The maximum radiation was 3804 W/m^2^ and the minimum was 0 W/m^2^. The THI during the experimental period averaged 73.4 ± 5.68 and the maximum and the minimum were 83.9 and 61.4, respectively.

### 3.2. Behavior

The treatments affected standing idle and grazing behavior (*p* = 0.0354 and *p* = 0.0435, respectively). Animals with access to water treatments (sprinklers and showers) spent more time standing idle than the animals provided with artificial shade (184, 188, and 156 min, respectively). Animals with access to showers spent more time grazing when compared to those with access to sprinklers and artificial shade (118, 109, and 108 min, respectively) (Table 2).

There was a breed effect on the behavioral activities of lying idle, ruminating, and grooming (*p* = 0.0006, <0.0001, and 0.0006, respectively) (Table 2). Jersey cows spent more time lying idle (64 vs. 53 min) and ruminating (111 vs. 109 min) during the hours observed than Holstein cows whereas the latter spent more time grooming than Jersey cows (28 vs. 23 min, respectively).

The age group and the interactions between treatment × breed and treatment × age group did not influence the behavioral patterns analyzed (*p* > 0.05).

### 3.3. Frequency and Duration of Use of the Systems

Significant differences were found concerning the frequency and duration of use of the systems by the animals (*p* < 0.0001) (Table 3). The animals with access to sprinklers and showers spent more time using these systems than the animals in the artificial shade group (184, 178, and 68 min, respectively), and consequently, the frequency of use of these resources was also higher (7, 7, and 1 times) (Figure 2).

The duration and frequency of use of the systems in were affected by the age group (*p* = 0.0004 and 0.013, respectively). Cows used (205 min) and sought out (7 times) the systems considerably more than the heifers (82 min and 3 times) (Table 3) (Figure 2).

Significant effects were found for the interaction between the treatment and age group on the duration of use (*p* < 0.0001). Cows used the sprinkler (270 min) and shower systems (268 min) for similar amounts of time, and for longer than heifers used the same systems (97 and 89 min, respectively) (Table 4). 

There was an interaction between treatment and age group on the frequency of use of the systems (*p* < 0.0001) (Table 4). Cows sought out the sprinklers (9 times) and showers (10 times) more times than heifers for the same systems (5 and 4 times, respectively) and showed a lower frequency of demand for shading, and this did not differ between breeds (1 time for cows and 1 time for heifers).

The time of day influenced the duration of use of the systems during the morning (9:00 to 11:00 a.m.), afternoon (12:00 to 3:00 p.m.), and late afternoon (4:00 to 6:00 p.m.) (*p* < 0.05) (Figure 3a–d). The animals used for the artificial shade system (59 min) for a longer period than the sprinkler system (35 min) and showers (21 min) during the morning; however, the frequency of use in the morning did not differ between treatments (*p* > 0.005) (Figure 3). During the afternoon, the animals used and sought out the sprinkler system (41 min and 2 times, respectively) and showers (51 min and 2 times, respectively) substantially more than the artificial shade system (6 min and 0.2 times, respectively). During the late afternoon, the animals used the sprinkler (37 min) and shower systems (32 min) more than the artificial shade system (2 min); however, the time of day did not influence the duration of use, being influenced only by treatments, since the sprinkler (2) and shower (2) treatments were sought out more often than the artificial shade system (0.1).

### 3.4. Physiological Parameters

RR and CS showed a high and significant correlation coefficient (*p* < 0.0001) with the THI (Figure 4a,b) in all treatments. The linear correlation coefficient between the THI and CS for artificial shading, sprinklers, and showers was R^2^ = 0.8572, R^2^ = 0.7886, and R^2^ = 0.6672, respectively. The linear correlation coefficient between THI and the RR was R^2^ = 0.7902 for artificial shading, R^2^ = 0.6286 for showers, and R^2^ = 0.6822 for sprinklers. RT did not show a significant correlation coefficient with THI (*p* < 0.0001) (Figure 4c). The linear correlation coefficient between the THI and RT for artificial shading, sprinklers, and showers was 0.1145, 0.4397, and 0.4354, respectively.

## 4. Discussion

### 4.1. Environmental Conditions

The air temperature averaged 26 ± 4.1 °C (mean ± standard deviation), a value considered above the thermal comfort zone for Holstein dairy cattle, which is 24 °C [15], but below the critical limit for Jersey cows (28 °C) [3]. The RH was 74 ± 11.3%, a value considered above the optimal range of 60 to 70% at which evaporative thermolysis between the animal and the environment is not harmed [16]. Due to the high air temperature and the relative humidity of the air, the THI values were high 73.4 ± 5.68, and considered harmful to animals, especially for high-yielding cows [17].

### 4.2. Behavioral Parameters 

Standing idle was the activity in which the animals spent most of their time (Table 2). The treatments that used water (sprinklers and showers) recorded the highest values of standing idle, a circumstance clarified by the fact that the animals were idle when using the systems in most cases.

The rumination and lying idle behaviors in the present experiment lasted the longest in Jersey cows. One of the factors that may be related to the longer ruminating time is the fact that these animals have a smaller mouth than Holsteins, so they spend longer grazing and consequently ruminating [18]. The increase in ruminating time and lying idle is also due to the greater adaptability of these animals to tropical climates, mainly due to their skin pigmentation, the size and density of their hair, and their higher sweating capacity, which gives them higher tolerance to heat [3]. Similarly, Aikman et al. [18] reported that Jersey cows spent more time grazing and ruminating than Holstein cows because they have smaller mouths, so they require a larger number of mouthfuls to ingest an equal volume of feed.

Grazing was the second most common activity. The animals exposed to the shower system spent the most time grazing. This may have been influenced by the benefits provided by the showers, including higher heat dissipation and a reduction in the coat surface temperature and respiratory rate (Figure 4a,b), bringing greater comfort to the animals and thus allowing them to graze for longer.

Grooming was the behavior performed the least frequently by animals and is characterized by contact, which can be coordinated or not, of short or long duration, initiated by the animal with objects, itself (the act of licking), or another animal [19]. Holstein cows performed grooming more frequently than Jersey cows, a fact that can be explained by the greater tendency to use the systems by the Holstein cows (*p* = 0.0876) (Table 3) since we observed that the behavior was mostly carried out on the iron structures where the systems were installed, showing that the systems also functioned as a form of environmental enrichment.

### 4.3. Frequency and Duration of Use of the Systems

When we analyzed the duration and frequency of use of the systems (systems + artificial shade) we found that the animals spent more time using and seeking out the sprinkler + artificial shade system (184 min) or the shower + artificial shade system (178 min) than the artificial shade system (68 min). This preference can be attributed to the greater efficiency in the dissipation of heat by evaporation, improving the welfare of the animals, especially those raised on pastures.

Figure 2 shows the comparison between the structures within the same paddock (sprinkling vs. artificial shading, in paddock 1; shower vs. artificial shading, in paddock 2; artificial shading in paddock 3). It is possible to perceive the preference for systems with water over artificial shade in the same paddock. This leads us to understand and respect the behavior and the expression of free choice of animals for systems that provide them with the most comfort and consequently best welfare. However, these behaviors varied according to the time of day (Figure 3a–d). The animals started using the systems at 9:00 a.m. (25 °C, 78% humidity, and 74.04 THI) with the frequency of use for sprinkler, shower, and artificial shade systems being similar, but with a longer duration of use of the artificial shade system until 11 a.m. (28.7 °C, 70.76% humidity, and 77.45 THI) when the temperature was cooler. After 11 a.m., the frequency and duration of use were higher for sprinkler and shower systems than for artificial shade, and this behavior continued until 4:00 p.m. (32 °C, 61% humidity, and 80.5 THI). It was also noticed that, from 2:00 p.m. to 3:00 p.m., the animals did not seek out the artificial shade system. This shows us the animals’ preference for systems that use water at times when the air temperature is higher. After 5:00 p.m. (32.04 °C, 61% humidity, and 79.2 THI), there was a decrease in the duration of use of the sprinkler and shower systems, but use remained higher than that of the artificial shade system. The demand for the systems ceased after 6:00 pm and there was no use overnight. 

The positioning of the animals in the systems presented specific characteristics. In the sprinkler system, the animals positioned themselves so that the droplets fell on most of their body, preferably on the back. In the shower system, the animals positioned themselves so that the water flow reached the flanks and pelvic region, with a preference for the left side, where the rumen is located, and where there would consequently be greater heat generation. In the artificial shade system, the animals presented stereotyped movements, stamping their hooves and moving their tails repeatedly in response to the water released by the sensors, so the use of artificial shade was also influenced as a false idea to trigger them.

There was also variation in the duration and frequency of use of treatments according to age group since cows used the systems for longer and sought them out more times than heifers. We believe there are two possible explanations for this. The first assumes that cows were dominant over heifers (submissive). Dominance in many cases is established by the competition for resources, often being the product of aggression among animals, and thus determining which animals will have access to the resource [20]. The second possibility comes from the fact that cows are more sensitive to heat than heifers since milk production increases metabolic heat [21].

### 4.4. Physiological Parameters

There was an increase in the coat surface temperature and respiratory rate with the THI in all treatments (Figure 4a,b). However, although increased, the coat surface temperature and respiratory rate in the sprinkler and shower treatments increased the least and were within the normal range. This shows that even though the THI was high and considered harmful for animals, the systems mitigated the adverse effects of the climate and the stress conditions, illustrating the benefits that the water systems brought to animal comfort and welfare [11,22], as water has a high caloric capacity and high latent heat of vaporization, thus decreasing elevated temperatures efficiently and favoring greater exchange of heat between the skin and the environment [10]. On the other hand, the use of such systems did not result in significant differences in RT, although there was a medium correlation between THI and RT for the shower systems (43.97%) and sprinkler (43.54%), and low correlations when the animals were submitted to the artificial shade system (11.45%). These values are within the range (38.1 to 39.1 °C) reported by West et al. [23]. This finding may be related to the greater heat dissipation of the animals [24,25] through the increase in RR, showing that all systems contributed to the maintenance of rectal temperature, with RT being one of the main indices for assessing the adaptability of animals [26].

## 5. Conclusions

In conclusion, dairy cows preferred to use sprinklers or showers over artificial shade in ambient conditions. These systems were more efficient at reducing the heat load and led to better behavioral and physiological parameters. The preference for shade over sprinklers or showers was more marked during the morning (9:00 a.m. to 11:00 a.m.), but as temperature, humidity and THI increased the preference for systems that use water was higher and almost exclusive, especially in the case of lactating and Holstein cows because they are more thermosensitive animals. Accordingly, it is recommended to use systems that use water to reduce heat stress, especially during the hot period of the year to improve the behavioral and physiological responses of animals, especially those more sensitive to heat. Considering the cost and the amount of water used and given the similar animal responses when using sprinkler vs. shower systems, it is recommended to use a sprinkler system to attenuate the impact of heat stress on dairy cattle.

## Figures and Tables

**Figure 1 animals-12-02318-f001:**
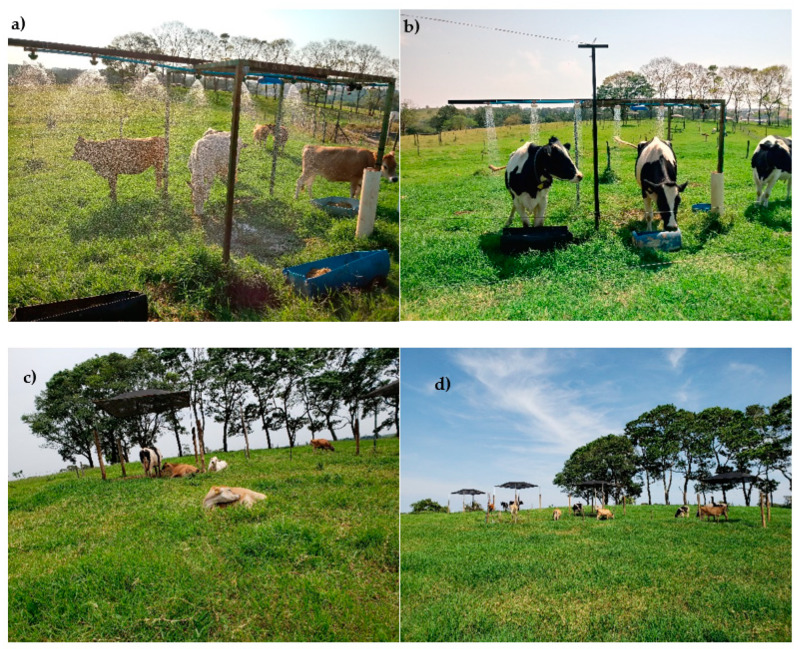
(**a**) Sprinkling + artificial shade; (**b**) showers + artificial shade; and (**c**) artificial shade. (**d**) All paddocks had the same artificial shading system at one end, comprising eucalyptus beams in which iron-based structures were fixed (3.5 m tall in total) with synthetic fabric (artificial shade) providing 80% solar retention. Source: Author (2018).

**Figure 2 animals-12-02318-f002:**
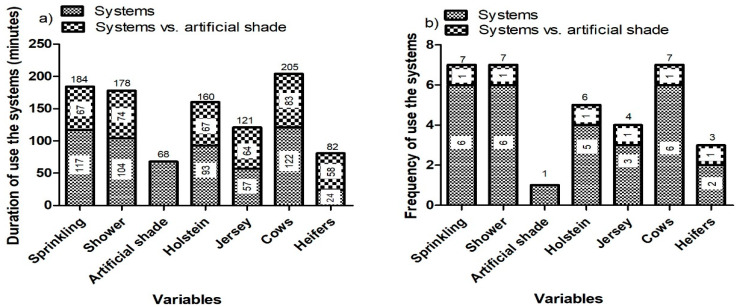
(**a**) Comparison between the structures of the same paddock in relation to duration of use of the systems: sprinkling vs. artificial shading; shower vs. artificial shading; and artificial shading; (**b**) Comparison between the structures of the same paddock in relation to frequency of use of the systems: sprinkling vs. artificial shading; shower vs. artificial shading; and artificial shading.

**Figure 3 animals-12-02318-f003:**
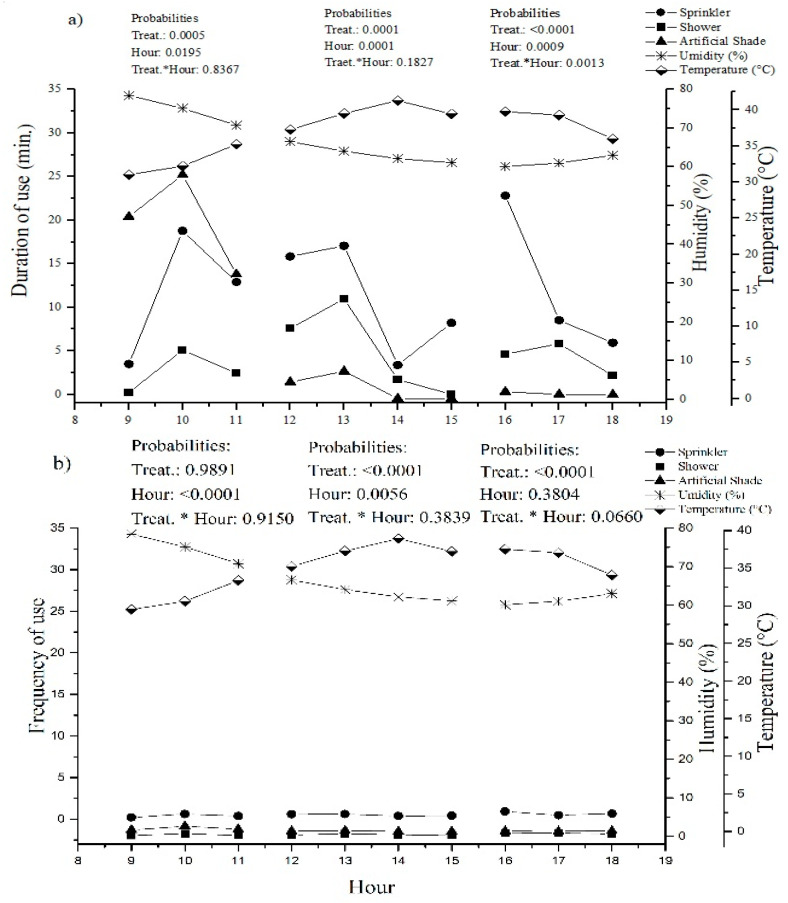
(**a**) Influence of duration of use (minutes) of sprinkler, shower, and artificial shade systems in relation to the hour of the day (*y* axes: duration of use, temperature, and humidity); (**b**) Influence of frequency of use of sprinkler, shower, and artificial shade systems in relation to the hour of the day (*y* axes: frequency of use, temperature, and humidity); (**c**) Influence of duration of use (minutes) of sprinkler, shower, and artificial shade systems in relation to the hour of the day (*y* axes: duration of use and THI); (**d**) Influence of frequency of use of sprinkler, shower, and artificial shade systems in relation to the hour of the day (frequency of use and THI).

**Figure 4 animals-12-02318-f004:**
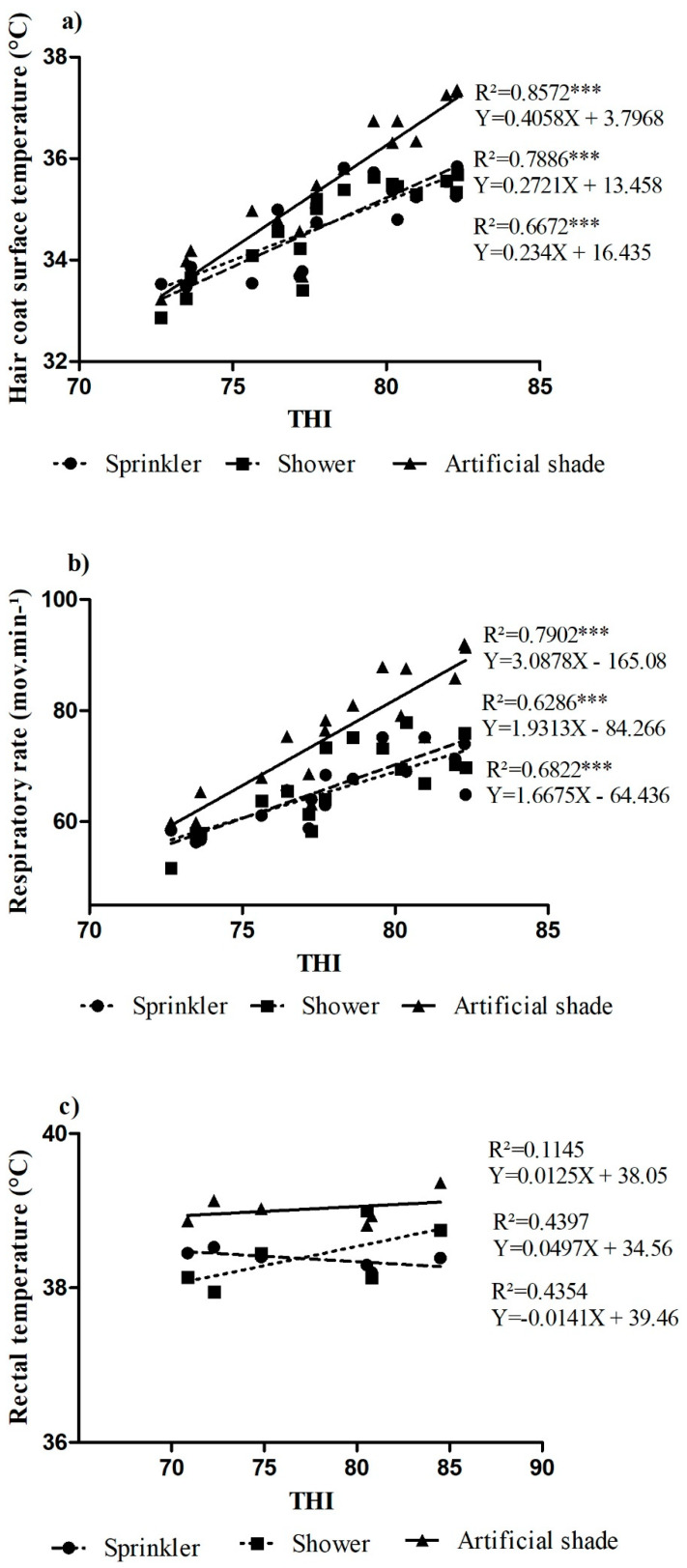
(**a**) Regression analysis of hair coat surface temperature (°C) of animals subjected to THI sprinkler + artificial shade, shower + artificial shade, and artificial shade systems; (**b**) Regression analysis of respiratory rate (mov. Min^−1^) of animals subjected to THI sprinkler + artificial shade, shower + artificial shade, and artificial shade systems. (**c**) Regression analysis of rectal temperature (°C) of animals subjected to THI sprinkler + artificial shade, shower + artificial shade, and artificial shade systems. Note: *** significant correlation coefficient (*p* < 0.0001).

**Table 1 animals-12-02318-t001:** Environmental conditions recorded during the experimental period.

Variables	Mean	Standard Deviation	Minimum	Maximum
Air Temperature (°C)	26	4.1	15.4	39.7
Relative Humidity (%)	74.07	11.3	41	94
Radiation (W/m^2^)	338.22	585.6	0	3804
THI ^1^	73.40	5.68	61.4	83.9

^1^ temperature and humidity index.

**Table 2 animals-12-02318-t002:** Duration (minutes) of behavioral activities in relation to different pasture heat-stress-reduction systems.

	Treatments			Breed		Age Group		S.E.M
	Sprinkler	Shower	As ^1^	Holstein	Jersey	Cows ^2^	Heifers ^3^	
Standing idle	184 ^a^	188 ^a^	156 ^b^	191	159	176	177	0.096
Lying idle	49	59	68	53 ^b^	64 ^a^	33	84	0.537
Grazing	109 ^b^	118 ^a^	108 ^b^	100	123	107	116	0.736
Ruminating	119	93	118	109 ^b^	111 ^a^	148	72	0.089
Grooming	20	25	30	28 ^a^	23 ^b^	17	34	0.344
*p* value								
	Treatment		Breed	Age group		Treat × Breed		Treat × Age group
Standing idle	0.0364		0.8845	0.1992		0.7009		0.9453
Lying idle	0.4256		0.0006	0.9968		0.9441		0.1581
Grazing	0.0435		0.4022	0.3989		0.3259		0.9145
Ruminating	0.7979		<0.0001	0.2037		0.3596		0.9451
Grooming	0.4341		0.0006	0.1829		0.6617		0.9123

^1^ artificial shade; ^2^ lactating cows; ^3^ pubescent heifers. Means followed by the same letter on the line do not differ from each other by the Tukey test, *p* < 0.05.

**Table 3 animals-12-02318-t003:** Frequency and duration (minutes) of use of the systems heat-stress-reduction systems.

	Treatments			Breed		Age Group		S.E.M
	Sprinkler	Shower	As ^1^	Holstein	Jersey	Cows ^2^	Heifers ^3^	
Duration of use	184 ^a^	178 ^a^	68 ^b^	160	121	205 ^a^	82 ^b^	11.062
Frequency of use	7 ^a^	7 ^a^	1 ^b^	6	4	7 ^a^	3 ^b^	0.280
*p* value								
	Treatment		Breed	Age group		Treat × Br ^4^		Treat × Ag ^5^
Duration of use	<0.0001		0.0876	0.0004		0.1849		<0.0001
Frequency of use	<0.0001		0.290	0.013		0.0566		<0.0001

^1^ artificial shade; ^2^ lactating cows; ^3^ pubescent heifers; ^4^ breed; ^5^ age group. Means followed by the same letter on the line do not differ from each other by the Tukey test, *p* <0.05.

**Table 4 animals-12-02318-t004:** Treatment × breed and treatment × age interactions on frequency and duration (minutes) of use of the systems heat-stress-reduction systems.

	Age Group					
	Pubescent Heifers			Lactating Cows		
	Sprinkler	Shower	Artificial Shade	Sprinkler	Shower	Artificial Shade
Duration of use	97 ^b^	89 ^b^	59 ^b^	270 ^a^	268 ^a^	76 ^b^
Frequency of use	5 ^B^	4 ^BC^	1 ^C^	9 ^A^	10 ^A^	1 ^C^

Means followed by the same letter on the line do not differ from each other by the Tukey test, *p* < 0.05. Upper case letters for treatment × age group interaction line, and lowercase letters for treatment × age interaction line.

## Data Availability

Data available by contacting Karoline L. Guimarães-Yamada or Geraldo T. dos Santos.

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
