# Peer review of "Influence of Different Heat-Stress-Reducing Systems on Physiological and Behavioral Responses and Social Dominance of Holstein and Jersey Cows and Heifers on Pasture"

_animals, 2022, doi:10.3390/ani12182318_

Round 1

Reviewer 1 Report

Review of Manuscript Animals-1873474

The paper aimed at evaluating the effect of different systems to reduce heat stress on cows and heifers of breeds Holstein and Jersey. Physiological parameters and the preferences were here evaluated. I have for the authors the following comments:

Major comments

Abstract is not giving all important information. Duration of experiment is missing, information about feeding and environmental conditions (e.g., temperature during experiment). When presenting results some numbers should be given to understand the magnitude of the effects. Finalize abstract with a general recommendation

Introduction shortly and clearly justified the realization of this study.

M&M were described with enough detail

Results were well and scientifically discussed and using substantial literature as support. I just wonder why authors did not report animal performance

Minor comments

L19: “that can improve the heat stress”? Probably you mean reduce

L32-35: Please give some number to better understand the effects and differences

L43: Weather is too general here. Be specific. Temperature?

L62: to be specific “lower” rectal temperature

L65: Was this small difference statistically significant? Otherwise, this statement is not admissible. Please check!

L88: The range of temperature could be also interesting to show. What about THI? Give the values

L90: What was the basis to select only 12 animals? The number is too low

L104: What kind of silage?

L135: in a and b I do not really see the natural shade

L146: Pity that rectal temperature was not considered. This is mostly used in such experiments and is normally reliable

L215: Also give the range of variation for all variables

L235: I think here the p value is <0.0001. And mention that there was a treatment effect to be specific

L241: I would recommend showing the exact p-values as in the table below. Check other cases

L265-276: I would recommend linking these results to temperature and especially to THI during the different periods of the day. Please show this value also in a Table like Table 1. Show also in figure 3, THI

L296-302: It is also important to mention range of values

L310: But they are also small. I think to compare it is important to considering the size of mouth in relation to the body size

L385-391: What are the implications and main recommendations for researchers and farmers based on this study?

Author Response

Reviewer # 1:

Abstract is not giving all important information. Duration of experiment is missing, information about feeding and environmental conditions (e.g., temperature during experiment). When presenting results some numbers should be given to understand the magnitude of the effects. Finalize abstract with a general recommendation

Reply: Yes, you are correct. Changes were made accordingly.

Abstract: High ambient temperatures and relative humidity affect the behavior and the physiology of the animal. This study investigated the influence of different heat-stress-reducing systems on the physiological, behavioral, and preferential responses of Holstein and Jersey cows and heifers on pasture. Experimental treatments were: 1) three heat-stress-reducing systems [sprinklers + artificial shade; showers + artificial shade; and artificial shade]; 2) two breeds (Holstein and Jersey); and 3) two physiological stages (lactating cows and pubertal heifers). Physiological and behavioral responses to treatments were measured every 30 minutes on collection days. Frequency and duration of the use of the systems were recorded continuously 24 h/day for 3 days in each period. The air temperature and the relative humidity averaged 26 ± 4.1°C and 74 ± 11.3%, respectively. The experimental treatments affected (P = 0.0354) standing idle, grazing behavior (P = 0.0435) and the frequency and the duration of use of the systems by the animals (P < 0.0001). For all treatments, the respiratory rate and the coat surface temperature were highly and significantly correlated (P < 0.05) with the temperature and humidity index. In conclusion, under ambient conditions, dairy cows preferred using sprinklers or showers over artificial shade. These systems were more efficient at reducing the heat load and led to better behavioral and physiological responses.

Results were well and scientifically discussed and using substantial literature as support. I just wonder why authors did not report animal performance.

Reply: This information was placed in another article in the process of being forwarded to another journal, where hormonal, mineral, reproductive, physiological, and productive factors were addressed. We don't think so much information could fit in a single article.

L19: “that can improve the heat stress”? Probably you mean reduce

Reply: Yes, you are correct, I meant reduce.

“This research shows the importance of using mechanisms that can reduce the heat stress of dairy animals raised on pasture.”

L32-35: Please give some number to better understand the effects and differences

Reply: Yes, you are correct.

High ambient temperatures and relative humidity affect the behavior and the physiology of the animal. This study investigated the influence of different heat-stress-reducing systems on the physiological, behavioral, and preferential responses of Holstein and Jersey cows and heifers on pasture. Experimental treatments were: 1) three heat-stress-reducing systems [sprinklers + artificial shade; showers + artificial shade; and artificial shade]; 2) two breeds (Holstein and Jersey); and 3) two physiological stages (lactating cows and pubertal heifers). Physiological and behavioral responses to treatments were measured every 30 minutes on collection days. Frequency and duration of the use of the systems were recorded continuously 24 h/day for 3 days in each period. The air temperature and the relative humidity averaged 26 ± 4.1°C and 74 ± 11.3%, respectively. The experimental treatments affected (P = 0.0354) standing idle, grazing behavior (P = 0.0435) and the frequency and the duration of use of the systems by the animals (P < 0.0001). For all treatments, the respiratory rate and the coat surface temperature were highly and significantly correlated (P < 0.05) with the temperature and humidity index. In conclusion, under ambient conditions, dairy cows preferred using sprinklers or showers over artificial shade. These systems were more efficient at reducing the heat load and led to better behavioral and physiological responses.

L43: Weather is too general here. Be specific. Temperature?

Reply: After the corrections, there were changes in the text above, shifting to Line - L 46-48:  High ambient temperatures, relative humidity, and solar radiation affects the behavior, physiology, and welfare of animals, especially in tropical countries during the summer.”

L62: to be specific “lower” rectal temperature

Reply: After the corrections, there were changes in the text above, shifting to Line - L-66:  “For example, Igono et al. [12] studied the use of sprinklers plus ventilation in the trough line on physiological and productive responses and concluded that animals that were sprinkled with more ventilation had lower rectal temperatures and produced 2 kg more milk per day when compared with animals with access to shade only.”

L65: Was this small difference statistically significant? Otherwise, this statement is not admissible. Please check!

Reply: Yes, that small difference was statistically significant, as you can see below

During the dry period, cows exposed to cooling had lower rectal temperatures in the morning (P < 0.001; Table 2) and afternoon (P < 0.001; Table 2) and a lower respiration rate (P < 0.001; Table 2) relative to cows in heat stress.

L88: The range of temperature could be also interesting to show. What about THI? Give the values

Reply:   After the corrections, there were changes in the text above, shifting to Line - L-90 - 95:

                According to the Köppen classification, the climate at Fazenda Experimental de Iguatemi is humid subtropical mesothermal (Cfa) with hot summers and rainfall concentrated in the summer months. During the experimental period, there was no rain and the mean temperature was 26°C ± 4.1°C while the temperature and humidity index (THI) was 73.4 ± 5.68. The maximum and the minimum environmental temperature recorded were 39.7°C and 15.4°C, respectively. The maximum and the minimum THI were 83.9 and 61.4, respectively (Table 1).

L90: What was the basis to select only 12 animals? The number is too low

Reply: The numbers of 12 animals seems to low but the use of the Latin square design and the factorial arrangement of the treatments increases the power of the test and detect significant differences between treatments as shown in the “Results section”. Thanks for the comment.

L104: What kind of silage?

Reply: After the corrections, there were changes in the text above, shifting to Line - L-115 “then fed with corn silage (8% crude protein and 50% neutral detergent fiber) and concentrate (26% crude protein and 10% neutral detergent fiber) and released back into the paddocks”

L135: in a and b I do not really see the natural shade

Reply: After the corrections, there were changes in the text above, shifting to Line - L-152: I changed the photos for a better view and conditioned added a fourth photo to show artificial shade from sprinkler systems and showers. As the artificial shading structures were at the end of the paddock, visualization is difficult in photos as they appear small at the end.

L146: Pity that rectal temperature was not considered. This is mostly used in such experiments and is normally reliable

Reply: After the corrections, there were changes in the text above, shifting to Line - L-171-173: We measured the rectal temperature, but we had left it for another article. Nevertheless, the rectal temperature data were added in the manuscript as requested.

L215: Also give the range of variation for all variables

Reply: After the corrections, there were changes in the text above, shifting to Line - L-236-247:  Ok

The environmental conditions recorded during the experimental period are shown in Table 1. The air temperature was 26 ± 4.1°C (mean ± standard deviation). The maxi-mum environmental temperature recorded was 39.7°C and the minimum temperature was 15.4°C. The relative humidity (RH) during the experimental period was 74 ± 11.3%, on average. The maximum and the minimum RH were 94% and 41%, respectively. The average radiation was 338.2 ± 585.6 (W/m²) and the THI was 73.4 ± 5.68. The maximum radiation was 3804 W/m² and the minimum was 0 W/m². The THI during the experimental period averaged 73.4 ± 5.68 and the maximum and the minimum were 83.9 and 61.4, respectively.

L235: I think here the p value is <0.0001. And mention that there was a treatment effect to be specific

Reply: After the corrections, there were changes in the text above, shifting to Line - L-272: Yes, you are correct. Thanks!

Significant differences were found for the frequency and duration of use of the systems by the animals (P<0.0001) (Table 3).

L241: I would recommend showing the exact p-values as in the table below. Check other cases

Reply: After the corrections, there were changes in the text above, shifting to Line - L-277: ok, I reviewed and fixed all that needed.

L265-276: I would recommend linking these results to temperature and especially to THI during the different periods of the day. Please show this value also in a Table like Table 1. Show also in figure 3, THI

Reply: After the corrections, there were changes in the text above, shifting to Line – 303-312: Ok, I do it.

However, these behaviors varied according to the time of day (Figure 3a, 3b, 3c and 3d). The animals started using the systems at 9:00 am (25 °C, 78% humidity and 74.04 THI) with the frequency of use for sprinkler, shower, and artificial shade systems being similar, but with a longer duration of use of the artificial shade system until 11 am (28.7 °C, 70.76% humidity and 77.45 THI) when the temperature was cooler. After 11 am, the frequency and duration of use was higher for sprinkler and shower systems than for artificial shade, and this behavior continued until 4:00 pm (32 °C, 61% humidity and 80.5 THI). It was also noticed that, from 2:00 pm to 3:00 pm, the animals did not seek out the artificial shade system. This shows us the animals' preference for systems that use water at times when the air temperature is higher. After 5:00 pm (32.04 °C, 61% humidity and 79.2 THI), there was a decrease in the duration of use of the sprinkler and shower systems, but use remained higher than that of the artificial shade system. The demand for the systems ceased after 6:00 pm and there was no use overnight.

L296-302: It is also important to mention range of values

Reply: After the corrections, there were changes in the text above, shifting to Line – L351-357: Ok, I do it.

The air temperature averaged 26 ± 4.1 °C (mean ± standard deviation), a value considered above the thermal comfort zone for Holstein dairy cattle, which is 24°C [15], but below the critical limit for Jersey cows (28°C) [3]. The RH was 74 ± 11.3%;, a value considered above the optimal range of 60 to 70% at which evaporative thermolysis between the animal and the environment is not harmed [16]. Due to the high air temperature and the relative humidity of the air, the THI values were high 73.4 ± 5.68 and considered harmful to animals, especially for high-yielding cows [17].

L310: But they are also small. I think to compare it is important to considering the size of mouth in relation to the body size

Reply: Ok

L385-391: What are the implications and main recommendations for researchers and farmers based on this study?

Reply: After the corrections, there were changes in the text above, shifting to Line – L 457-469:

Ok, done. Thanks

In conclusion, dairy cows preferred to use sprinklers or showers over artificial shade in ambient conditions. These systems were more efficient at reducing the heat load and led to better behavioral and physiological parameters. The preference for shade over sprinklers or showers was more marked during the morning (9:00 am to 11:00 am), but as temperature, humidity and THI increased the preference for systems that use water was higher and almost exclusive, especially in the case of lactating and Holstein cows because they are more thermosensitive animals. Accordingly, it is recommended to use systems that use water to reduce heat stress, especially during the hot period of the year to improve behavioral and physiological responses of animals, especially those more sensitive to heat. Considering the cost and the amount of water used, and given the similar animal responses when using sprinkler vs. shower systems, it is recommended to use sprinkler system to attenuate the impact of heat stress on dairy cattle

We are immensely grateful for the suggestions and corrections of Reviewer #1 and of all suggestions we try to respond as much as possible.

Reviewer 2 Report

TITLE: INFLUENCE OF DIFFERENT HEAT-STRESS-REDUCING SYSTEMS ON PHYSIOLOGICAL AND BEHAVIORAL RESPONSES AND SOCIAL DOMINANCE OF HOLSTEIN AND JERSEY COWS AND HEIFERS ON PASTURE.

GENERAL COMMENTS

The subject of this article is very interesting that should be of interest to Animals mdpi readers. Organisation of the sections and structure are appropriate. Overall, the article is good, and I only have just minor/specific comments and suggestions to the authors.

SPECIFIC COMMENTS

[lines 81-82] Please, you should indicate the days.

[line 88] Please, you should also indicate the minimum and the maximum temperature.

[Figure 1] The photos in figure 1 look very warped, can you put them right?

[line 148, 154] “…from 9 am to 6:30 pm…”. Please, change 9 to 9:00.

[References] Please, you should revise the format style of references of the whole article.

[line 397] this…This, capital letter.

ENGLISH LANGUAGE AND STYLE

English language and style are fine/minor spell check required.

OVERALL RECOMMENDATION

Accept after minor revision.

Author Response

Reviewer # 2:

[lines 81-82] Please, you should indicate the days.

Reply: After the corrections, there were changes in the text above, shifting to Line – L 85-88:

The experiment was carried out from the 10th of October to the 21th of November 2018 (Southern Hemisphere spring) at Fazenda Experimental de Iguatemi (FEI), a property belonging to the State University of Maringá (UEM), located at latitude 23º 25’ S, longitude 51º 57’ W, and 550 meters of altitude.

[line 88] Please, you should also indicate the minimum and the maximum temperature.

Reply: After the corrections, there were changes in the text above, shifting to Line – L 90-95:

             According to the Köppen classification, the climate at Fazenda Experimental de Iguatemi is humid subtropical mesothermal (Cfa) with hot summers and rainfall concentrated in the summer months. During the experimental period, there was no rain and the mean temperature was 26°C ± 4.1°C while the temperature and humidity index (THI) was 73.4 ± 5.68. The maximum and the minimum environmental temperature recorded were 39.7°C and 15.4°C, respectively. The maximum and the minimum THI were 83.9 and 61.4, respectively (Table 1).

Table 1. Environmental conditions recored during the experimental period.

Variables

Mean

Standard deviation

Minimum

Maximum

Air Temperature (°C)

26

4.1

15.4

39.7

Relative Humidity (%)

74.07

11.3

41

94

Radiation (W/m²)

338.22

585.6

0

3804

THI¹

73.40

5.68

61.4

83.9

¹ temperature and humidity index.

L-135: [Figure 1] The photos in figure 1 look very warped, can you put them, right?

Reply: After the corrections, there were changes in the text above, shifting to Line – L 152:

Ok. I changed the photos for a better view and conditioned added a fourth photo to show artificial shade from sprinkler systems and showers. As the artificial shading structures were at the end of the paddock, visualization is difficult in photos as they appear small at the end.

[line 148, 154] “...from 9 am to 6:30 pm...”. Please, change 9 to 9:00.

Reply: After the corrections, there were changes in the text above, shifting to Line – L171-173:

 “were measured every 30 minutes on the collection days from 9:00 am to 6:30 pm, except at the time of milking, in all animals”

“The behavioral patterns of each animal were analyzed every half hour on the sample collection days from 9:00 am to 6:30 pm, except”

[References] Please, you should revise the format style of references of the whole article.

Reply: There really were some wrong references, I already fixed them.

[line 397] this...This, capital letter.

Reply: “This study was funded in part by….”

We are immensely grateful for the suggestions and corrections of Reviewer #2 and of all suggestions we try to respond as much as possible.

Round 2

Reviewer 1 Report

I have read the revised version of the manuscript that aimed to evaluate the effect of different systems to reduce heat stress on cows and heifers of breeds Holstein and Jersey. Thanks a lot to the authors for the efforts invested on improving the quality of the paper and for responding in detail all my questions. I have no more remarks to the paper